# Dynamic Imaging of IEL-IEC Co-Cultures Allows for Quantification of CD103-Dependent T Cell Migration

**DOI:** 10.3390/ijms22105148

**Published:** 2021-05-13

**Authors:** Karin Enderle, Martin Dinkel, Eva-Maria Spath, Benjamin Schmid, Sebastian Zundler, Philipp Tripal, Markus F. Neurath, Kai Hildner, Clemens Neufert

**Affiliations:** 1Department of Medicine 1, Kussmaul Campus for Medical Research Universitätsklinikum Erlangen, University of Erlangen-Nuremberg, 91052 Erlangen, Germany; Karin.Enderle@uk-erlangen.de (K.E.); Martin.Dinkel@uk-erlangen.de (M.D.); Eva-Maria.Spath@uk-erlangen.de (E.-M.S.); Sebastian.Zundler@uk-erlangen.de (S.Z.); Markus.Neurath@uk-erlangen.de (M.F.N.); 2Deutsches Zentrum Immuntherapie (DZI), Universitätsklinikum Erlangen, University of Erlangen-Nuremberg, 91054 Erlangen, Germany; 3Optical Imaging Centre Erlangen (OICE), University of Erlangen-Nuremberg, 91052 Erlangen, Germany; benjamin.schmid@fau.de (B.S.); philipp.tripal@fau.de (P.T.)

**Keywords:** intraepithelial lymphocytes, intestinal organoids, life cell imaging, CD103, lymphocyte migration

## Abstract

Intraepithelial lymphocytes (IEL) are widely distributed within the small intestinal epithelial cell (IEC) layer and represent one of the largest T cell pools of the body. While implicated in the pathogenesis of intestinal inflammation, detailed insight especially into the cellular cross-talk between IELs and IECs is largely missing in part due to lacking methodologies to monitor this interaction. To overcome this shortcoming, we employed and validated a murine IEL-IEC (organoids) ex vivo co-culture model system. Using livecell imaging we established a protocol to visualize and quantify the spatio-temporal migratory behavior of IELs within organoids over time. Applying this methodology, we found that IELs lacking CD103 (i.e., integrin alpha E, ITGAE) surface expression usually functioning as a retention receptor for IELs through binding to E-cadherin (CD324) expressing IECs displayed aberrant mobility and migration patterns. Specifically, CD103 deficiency affected the ability of IELs to migrate and reduced their speed during crawling within organoids. In summary, we report a new technology to monitor and quantitatively assess especially migratory characteristics of IELs communicating with IEC ex vivo. This approach is hence readily applicable to study the effects of targeted therapeutic interventions on IEL-IEC cross-talk.

## 1. Introduction

Balance of the multi-faceted epithelia-microbiota-immune communication is considered to be central to maintain intestinal homeostasis in healthy individuals [1]. Under pathogenic conditions of the intestines as e.g., in inflammatory bowel disease (IBD) there is accumulating evidence suggesting that both epithelia-microbiota, as well as epithelia-immune cell interaction, are dysregulated putatively driving disease manifestation and/or contributing to its progression [1,2]. The vast majority of immune cells residing in and interacting with intestinal epithelial cells (IEC) belong to the subset of lymphocytes that are hence denominated as intraepithelial lymphocytes (IEL) [3,4,5]. In stark contrast to subepithelially positioned lamina propria T cells that are predominately composed of the T cell receptor (TCR) alpha beta+ CD4+ T helper cells, small intestinal IELs consist of both TCR alpha beta+ and TCR gamma delta+ subpopulations with CD8+ IELs clearly outnumbering CD4+ T cells [6]. Developmentally, IELs can be categorized into so-called naturally occurring subsets that express either TCR gamma delta or TCR alpha beta clonotypes and enter the gut early in life in a largely microbial signal-independent manner [4,5]. In contrast, so-called induced IELs are exclusively TCR alpha beta+ and are generated postnatally upon immune-microbial interaction presumably resulting from a local tissue injury incidence (e.g., infection) breaching the intestinal barrier integrity [4,5]. Hence, within an individual and at a given anatomic site, composition, TCR repertoire, and functional abilities of especially induced IEL subsets are assumed to be unique and distinct thereby reflecting the intrinsic ability of IELs to adopt and adapt to the local tissue microenvironment [6,7]. Consequently, small intestinal IELs usually are memory T cells that have been lately identified to belong to the pool of tissue-resident memory (TRM) T cells that are characterized by CD69, CD44 and for the vast majority of IELs by CD103 (syn. integrin alpha E, ITGAE) expression on the cell surface [8].

Generally, IELs display both adaptive and innate immune functions and are overall assumed to exert context-dependent barrier-protective (e.g., prevention and control of infections, tissue repair or immune-regulation) or barrier-disruptive effects (e.g., during pathogen invasion or in celiac disease) [9,10,11]. Due to their intimate localization inside the intestinal epithelial layer in close proximity to the microbiota commonly known to strongly impact intestinal homeostasis, dysregulated IEL responses and hence consecutively altered IEL-IEC cross-talk might have a significant impact on the pathogenesis of IBD [12]. In line with a functional contribution of CD8+ T cells, both data from a murine ileitis model system as well as clinical, histopathological, and molecular data from patient cohorts imply that cytotoxic T cell responses are of pathogenic importance in IBD [13,14,15]. However, studies providing further proof of evidence for this concept are limited at least in part due to the absence of suitable experimental systems and methodologies to gain novel insight into the immunological, metabolic, and kinetic properties of IELs ex vivo in the context of IEC.

Seminal work discovering intestinal epithelial Lgr5+ stem cells within intestinal crypts as the crucial precursor paved the way for the broad availability of protocols to differentiate stem cells into three-dimensional intestinal organoids via the intermediate state of sphere-shaped intestinal epithelial cell structures ex vivo [16,17]. Moreover, the option of intestinal organoid generation ex vivo opened up the possibility to overcome the biggest obstacle of IEL-centered studies, i.e., the lack of protocols to reliably culture and study IELs ex vivo over time. In fact, small intestinal organoids were consecutively shown to provide the proper microenvironment allowing for IEL cultivation in a co-culture system ex vivo [18,19].

Given numerous, unresolved questions in regard to the cellular, molecular and functional characteristics of IEL-IEC cross-talk, in this study, we wanted to establish and functionally validate a protocol that allows us to assess the behavioral properties of small intestinal IELs within intestinal organoids ex vivo. Here, we provide a methodology to kinetically perform live-cell imaging of murine IELs within intestinal organoids to characterize and quantitate IEL migration behavior ex vivo. Applying this protocol, we found that the normal migratory behavior of IELs within E-cadherin expressing organoids is strongly dependent on CD103 surface expression and that the velocity of CD103 deficient IELs migrating through the organoid was significantly reduced. Hence, our studies provide a readily applicable method that allows for rapid experimental assessment of IEL behavior in the context of its natural habitat, the intestinal epithelium, in a spatio-temporal manner and functionally upon targeted intervention.

## 2. Results

### 2.1. Experimental Ex Vivo Co-Culture System Resembles the Spatial Distribution of Intraepithelial Lymphocytes within the Intestinal Epithelial Cell Layer

It is complicated to study IEL biology ex vivo since IELs are fragile and short-living when they were removed from their natural surroundings, i.e., the intestinal epithelial layer. To overcome this challenge, we took advantage of an ex vivo co-culture model that was recently described [18]. In fact, we purified intestinal crypts containing stem cells from the small intestine of mice and cultured them in matrigel ex vivo using standard procedures [17]. After 2 days, we purified IELs from the small intestine and initiated co-cultures in the presence of a cytokine cocktail including IL-2, IL-7, and IL-15 enabling growth conditions that allowed the ex vivo culture of IELs for more than a week. The experimental setup is illustrated in Figure 1A. For direct comparison with the situation in vivo, we analyzed cryosections from the ileal tissue of wildtype mice. Cross-sections were stained by immunofluorescently labeled antibodies for CD3 and EpCAM demonstrating co-localization of IELs and IECs along the crypt-villus axis by confocal microscopy (Figure 1B). Next, we cultured IELs in the presence of IEC organoids (as described in Figure 1A) before co-cultures were embedded in paraffin, stained by IF, and analyzed by confocal microscopy (Figure 1C) showing that IELs localize in close contact with the epithelial layer of IEC organoids. Then, co-cultures from corresponding setups were studied by spinning disc microscopy and 3D reconstruction further highlighting the spatial distribution of IELs within the IEC layer in our experimental model (Figure 1D). Whereas the imaging hitherto was performed on fixed cells and tissue, we next intended to clarify whether co-cultures of IELs and small intestinal are also suited to be studied by live cell imaging. Technically, fixed cells and tissue are eligible for antibody-based stainings, e.g., anti-CD3. By contrast, live cell studies require the use of live cell dyes. Hence, live cell imaging of IEL-IEC co-cultures was performed with fluorescently pre-stained IELs using a cell proliferation dye. Strikingly, confocal microscopy in combination with differential interference contrast (DIC) imaging enabled us to visualize such living co-cultures (Figure 1E). To further confirm the presence of IELs within the epithelial layer, live cell imaging and 3D reconstruction were also performed with co-cultures grown in matrigel stained with Lucifer Yellow (Figure 1F). The results of this approach were in line with the previous findings, showing that our ex vivo co-culture system resembles closely the in vivo constellation.

### 2.2. Intraepithelial Lymphocytes Stay Alive, Phenotypically Stable and Localize to the Intestinal Epithelial Cell Layer during Ex Vivo Co-Culture

Whereas previous experiments demonstrated the proof-of-principle of our setup, we then sought to address the feasibility of growing and monitoring such IEL-IEC co-cultures over several days. Here, we observed that the system is applicable showing live IELs localizing and moving within the IEC layer during culture and expansion of IEL-IEC co-cultures during one week as evaluated by confocal microscopy in combination with differential interference contrast (DIC) imaging (Figure 2A).

For our ex vivo setup, we used freshly isolated IELs and the purification included a MACS–based CD3+ enrichment. First, we directly assessed the composition of the pool of CD8α+ and CD4+ IELs at the beginning of the co-culture. Performing FACS analysis for CD4 and CD8α we observed that more than two-thirds of CD3+ IELs were CD4-CD8α+ (Figure 2B,C). Subsequent analysis of IELs obtained from the co-culture setup at days 2, 5, 7 revealed that the CD4-CD8α+ subpopulation remained at a similarly high level as compared to d0 and represented the subset with the vast majority of cells within the IEL compartment throughout the analyses. Both the CD4 + CD8α− and CD4 + CD8α+ cells were present at markedly lower frequencies, but they displayed specific subsets that were clearly detectable at each point of time (Figure 2B,C). Moreover, the vast majority of cells within the IEL compartment stained positive for CD69 and CD103 during serial analysis thus confirming the presence and stable phenotype of classic cell surface markers of IELs/TRMs in our co-culture system.

Thus, our data strongly suggest that the composition of IELs remains rather stable for one week during IEL-IEC co-culture, although a low degree of preferential expansion or cell death within IEL subsets cannot be excluded (Figure 2B,C).

Interestingly, in comparative studies using CD3+ splenic T cells or IELs in co-cultures with IEC organoids, we detected that splenic T cells were inefficient to localize to and to move within the IEC layer suggesting striking biological differences between splenic T cells and IELs also in this regard (Figure 2D).

In sum, our data strongly suggest that IECs stay alive, phenotypically stable, and localize to the IEC layer during our vitro co-culture conditions during a period of at least one week.

### 2.3. The Mobility of Intraepithelial Lymphocytes Can Be Monitored by Life Cell Imaging and Quantified by Track Analysis in Time-Lapse Studies

When we performed live cell imaging using spinning disc microscopy and DIC imaging, we observed that many of the IELs did not remain statically at a certain location, but shifted positions over time indicating that they were moving (Appendix A: IEL-IEC_co-culture_timelapse). Notably, the vast majority of movements occurred within the IEC layer. Next, we wanted to analyze the cell flow/mobility systematically. Therefore, we monitored the cell migration closely using time-lapse studies. Here, individual cells were tracked and all positional changes were followed over a defined period of time (Figure 3A). Thereby, we established a methodology to image and measure IELs movements as exemplified by color coding of four representative IELs and their corresponding track lines (Figure 3A). Strikingly, summation of all IEL tracks detectable resulted in the visualization of the global IEL movement patterns demonstrating high IEL mobility and suggesting that the IEC layer is patrolled by IELs thoroughly (Figure 3B).

Thus, our work indicates that life cell imaging and track analysis in time-lapse studies is suited to characterize global IEL movement patterns in the IEL-IEC co-culture model.

### 2.4. CD103-Deficient Intraepithelial Lymphocytes Display Aberrant Mobility and Migration Patterns

Based on our observations that IELs retained high expression of CD103 (which is known to interact with E-cadherin/CD324 on IECs) and demonstrated intense mobility within the IEC layer, we hypothesized that in our co-culture model system setup IEL movement patterns within IEC organoids could be dependent on CD103.

To test this hypothesis, we planned to compare the migratory abilities of CD103 sufficient (wildtype/WT) and CD103 deficient (CD103^−/−^) IELs using IELs from CD103 targeted mice [20]. We performed live cell imaging by spinning disc confocal microscopy/DIC imaging and cell flow was monitored using track analysis in time-lapse studies as described above. Strikingly, we noticed that the movement of CD103^−/−^ IELs within the IEC layer was substantially compromised as compared to wildtype IELs (Figure 4A). To evaluate whether these alterations displayed a global feature of CD103^−/−^ IELs or whether they were related to the direct contact with organoids, we tracked the movements of IELs not only in projection to IECs (inside tracks) but also outside the organoids (outside tracks) (Figure 4A). For a detailed analysis, we quantified the number of tracks in multiple IEL-IEC co-cultures. Interestingly, we observed that the number of all tracks was dramatically reduced in co-culture systems containing *CD103^−/−^* IELs as compared to wildtype IELs (Figure 4B). Of note, whereas the number of inside tracks was highly significantly reduced in CD103^−/−^ IELs, the number of outside tracks was comparable between both groups suggesting that the differential changes were caused rather by a defective interaction with IECs than by any CD103^−/−^ IEL genotype intrinsic defect in IEL mobility. In line with that, the number of inside tracks was significantly higher than the number of outside tracks for wildtype IELs, but not for CD103^−/−^ IELs (Figure 4B). Correspondingly, we did not obtain evidence for any major difference in transition events between wildtype and CD103^−/−^ IELs (Figure 4C). Next, we sought to quantify the speed of the IEL movements in the co-culture model. Accordingly, we calculated the speed of IEL movements based on the length of tracks in our time-lapse studies. To adjust for limitations resulting from the necessary transformation of a 3D model into 2D measurements, we decided to focus on tracks being suggestive of occurring at the nearly horizontal level during confocal microscopy. Notably, we observed that the speed of CD103^−/−^ IELs was significantly lower as compared to wildtype IELs (Figure 4D). Consistently, this interesting difference was only present when inside tracks were studied. By contrast, it was not detected during outside track analysis providing further evidence that the alteration was rather the consequence of a defective interaction between IELs and IECs than by an intrinsic loss in IEL mobility (Figure 4D). In addition, whereas the speed of wildtype IELs was similar inside and outside the IEC organoid, the speed of CD103^−/−^ IELs was reduced inside as compared to outside (Figure 4D).

To further characterize the mobility patterns of IELs in our ex vivo co-culture model, we quantified the longest tracks indicating maximum displacement. Interestingly, we found that the maximum displacement was highly significantly reduced for CD103^−/−^ IELs as compared to wildtype IELs (Figure 4E). Again, this remarkable finding of reduced maximum displacement of CD103^−/−^ compared to wildtype IELs was only found when IELs were tracked inside but not outside the organoids (Figure 4E). Moreover, the maximum displacement of wildtype IELs in our model was higher inside as compared to outside the IEC layer, but very similar when the maximum displacement was compared between inside and outside for CD103^−/−^ IELs (Figure 4E).

Thus, our data strongly suggest that CD103-deficient IELs exhibit defective mobility and migration patterns within organoids that can be measured by live cell analysis in an IEL-IEC co-culture model ex vivo.

## 3. Discussion

Tissue imprinting of both immune and epithelial cells is assumed to represent a continuously occurring and dynamic process during which a plethora of signals derived from their mutual interaction, stromal cells, and microbiota (including putative pathogens) are integrated [7,21]. This process continuously shapes the balance between immune and epithelial cell compartments while recurring challenges, e.g., by pathogenic environmental (i.e., food- or microbiota-related) signals, constantly put this equilibrium at risk. Due to their anatomic localization and tremendously plastic functionality alongside the intestinal tract, IELs sine qua non need to go through a vigorous process of tissue adaptation to properly act in a context-dependent manner. This can occur either as crucial immune-regulatory gatekeepers of intestinal homeostasis or—in case of tissue injury and danger—as pro-inflammatory guards that step up, attack, and then ideally resolve tissue inflammation resulting in healing and reconstitution of the intestinal barrier. However, detailed insights into the role of IELs in this complex system of “checks and balances” of the epithelial-microbiota-immune communication warranting intestinal homeostasis on the one hand and IELs functional contribution to T cell-mediated tissue inflammation in IBD on the other hand remain obscure. This could be largely due to the lack of informative model systems suitable to elucidate characteristics, behavior, and mode of action of IELs in the context of their natural environment, the intestinal epithelium.

Hence, in this study, we sought to develop applications to visualize and functionally study IEL biology in a quantitative manner ex vivo. For this, we adopted a previously published IEL–IEC co-culture model using ex vivo generated intestinal organoids and small intestinal IELs in the presence of T cell growth supportive cytokine mix [18]. Importantly, IEL-IEC co-cultures represent a major technical advance overcoming long-existing issues with culturing viable IELs alone in vitro, although some benefit (i.e., increased IEL viability and phenotype stability ex vivo) was reported upon addition of recombinant cytokine cocktails before [22,23]. However, prolonged culture times of IELs in vitro (without their IEC counterpart) reportedly induce significant changes in the cellular composition and putatively also the molecular profile of IELs thereby limiting the potential value of subsequent analyses [23]. Furthermore, IEC-derived signals regulating and shaping tissue imprinting and adaptation of IELs are only found in co-culture models further underscoring their exceptional advantages in this regard. Strikingly, we found that the employed IEL-IEC co-culture system allowed the preservation of viable IELs and TRM phenotype over the time span of at least one week which was similar to a previous report [18]. Interestingly, we could extend current knowledge by showing that IELs compared to splenic T cells are in fact superior in regard to their migratory properties within organoids indicating that IELs have encountered tissue site-specific abilities enabling IELs to specifically patrol through IEC structures. Our data should be interpreted in the context of previous work demonstrating that peripheral T cells can acquire features of IELs upon activation in the presence of enteroids [19].

Regardless, the value of IEL-IEC co-cultures is also limited since additional cell types (e.g., fibroblasts or other stromal cells) and environmental signals (e.g., microbiota) are usually found in the intestines with significant impact on IEL-IEC biology are missing in this experimental setup. In particular, environmental cues could play a critical role during CD8+ T cell migration in the small intestine [24]. Future studies should hence seek to design and establish more complex model systems that contain also stromal cells and allow “colonization” with commensals in an ex vivo setting thereby reflecting best the complex multifaceted epithelial-immune-microbiota communication taking place in vivo.

Employing the methodology of IEL-IEC co-cultures, we confirmed by various means that in the chosen ex vivo setup IELs, in fact, re-organize along the IEC layer within the organoids phenotypically recapitulating respective IEL behavior in vivo. Extending data sets displaying “snap-shot”-like pictures generated by confocal microscopy of fixed material, live cell imaging was established allowing for real-time assessment of IEL localization, mobility, and overall migration pattern in a living IEL-IEC co-culture setting. In addition, we set up time-lapse studies and developed specific tools including software scripts that enabled us to identify, mark and track IELs and their movement over time on a per cell level which might further promote the broad usage of this model. Thus, this protocol can serve to gain unique insights into the mobility and functional behavior of the global IEL population at a given time. In addition, it also provides the possibility to track single-cell fates over time. Moreover, whereas the present study focused on analyses of the IEL population as a whole, future studies might also address specific distributional or migratory characteristics of IEL subsets.

To validate our panel of analysis tools on IEL motility within IEC organoids ex vivo, we decided to assess the impact of CD103 (expressed on most small intestinal IELs) on the migratory behavior of IELs in the IEL- IEC co-culture model system. Originally, CD103 was discovered to be expressed on especially intraepithelial T cells while over the decades of research also non-IELs (e.g., regulatory T cells) and non-T cells (e.g., dendritic cells) have been identified to express CD103 [25]. The ligand of CD103 discovered so far represents E-cadherin that is predominately expressed by epithelial cells including IECs implying that CD103 is closely related to IEC biology [26]. In line with this, CD103 deficient mice display reduced IEL numbers that are reportedly not due to defective homing but rather due to hampered retention within the epithelial cell compartment suggesting that CD103 majorly plays a role as a retention receptor [20,25]. Functionally, CD103 was shown to be critically required for the phenotypical generation of TRM T cells in the small intestine that was dependent on TGF-β [27]. Furthermore, in the context of intestinal inflammation, T cells deficient in CD103 were compromised in inducing intestinal Graft-versus-Host disease (GvHD) upon allogeneic hematopoietic stem cell transplantation with hampered killing and/or retention within the intestinal epithelium reportedly representing the most likely underlying mechanism [28]. Finally, clinical studies testing etrolizumab—a monoclonal antibody blocking beta 7 integrin, the heterodimeric partner of CD103—in IBD are currently underway providing putatively additional indirect evidence that CD103 may represent a novel promising therapeutic target candidate in the future [29].

Interestingly, assessing CD103 deficiency in our model system revealed that the mobility and speed of IELs occur in a CD103-dependent manner. That result might appear counterintuitive to a certain extent since—as discussed above—CD103 is assumed to mediate adhesion to E-cadherin expressing epithelial cell structures. However, although we cannot provide mechanistic data clarifying this rather unexpected finding so far, it is tempting to speculate about potential reasons. A possible explanation could be that rather than acting like border patrol stations serving the purpose to slow down traffic, conversely, CD103/E-cadherin interaction might function similar to a rope ladder in mountain climbing: providing guidance to IELs and enabling them to patrol through IEC layers in a fast and efficient manner. However, future studies need to ultimately examine the underlying mechanism and the molecular and functional consequences for the altered IEL-IEC cross-talk upon blocked CD103/E-cadherin interaction.

In summary, the methodology reported here potentially may represent a novel platform to efficiently assess the impact of a given genetic alteration and/or targeted treatment on the IEL-IEC cross-talk ex vivo thereby putatively reserving in vivo testing with more stressful measures as e.g., in vivo intravital imaging to those cases that showed a meaningful signal in this ex vivo co-culture model system. Hence, currently reported tools assessing IEL biology in vivo could nicely complement our protocol [30].

Overall, IEL-IEC co-cultures and structured quantification of IEL migratory behavior ex vivo represent an achievable, cost- and time-effective tool to visualize and assess the functional outcome of targeted interventions of IEL-IEC cross-talk ex vivo.

## 4. Materials and Methods

### 4.1. Mice

Cells were purified for ex vivo studies from 10–12 weeks old mice housed in individually ventilated cages under SPF conditions at the animal facilities of Präklinisches Experimentelles Tierzentrum (PETZ) or First department of Medicine, Universitätsklinikum Erlangen. CD103 knockout mice (B6.129S2(C)-Itgae<tm1Cmp>) were described previously [20]. T cell reporter mice were generated by crossing CD4 Cre mice [31] with tdTOMATO flox mice [32]. C57BL/6J mice were used as controls. This study was carried out in accordance with the current legislation and the guidelines of the government of Lower Franconia in Bavaria, Germany.

### 4.2. Organoid Culture

To isolate crypts for small intestinal organoid culture, mice were sacrificed and the small intestine was removed. The gut was rinsed with cold PBS and cut into four to five pieces. These were cut longitudinal and villi were scratched off to expose the crypt region. In 0.5 cm small cut pieces were incubated in PBS containing 2 mM EDTA on a rotator at 4 °C for 30 min. Tissue pieces were transferred to 10 mL PBS, vortexed for 15 s, and put on top of a 70 µm filter. This was repeated four times. Flowthrough was spun down and washed with 10 mL basal culture media (BCM: DMEM Advanced F12 (Life Technologies, Carlsbad, CA, USA), 1% Penicillin-Streptomycin (Sigma, St. Louis, MI, USA), 1% HEPES (Sigma, St. Louis, MI, USA), and 1% Glutamax (Life Technologies, Carlsbad, CA, USA)) [17]. Seeding density was calculated based on the number of crypts and 250 crypts in 50 µL 1:1 BCM-matrigel mix (Matrigel from Corning, New York, NY, USA) were plated in a Nunclon Delta Surface plate (Thermofisher, Waltham, MA, USA). Next, incubation was performed at 37 °C for 10 min to induce matrigel polymerization followed by addition of complete culture media (CCM, 10% R-Spondin (culture supernatant from R-Spo1 producing cell line), 1× B27 supplement (Life Technologies, Carlsbad, CA, USA), 1mM N-Acetyl-L-Cysteine (Sigma, St. Louis, MI, USA), 100 ng/mL Noggin (Peprotech, Rocky Hill, NJ, USA), 20 ng/mL EGF (Immunotools, Friesoythe, Germany).

### 4.3. Isolation of Intraepithelial Lymphocytes

After mice were sacrificed, small intestines were immediately placed in cold PBS. Fat and Peyer’s Patches were removed and the intestines were rinsed with cold PBS. After longitudinally opening of the intestines, they were cut into small pieces and put into 20 mL HBSS (Sigma, St. Louis, MI, USA) supplemented with 5% FSC (Pan Biotech, Aidenbach, Germany), 10 mM HEPES (Sigma, St. Louis, MI, USA), 5 mM EDTA and 1 mM Dithiothreitol (Sigma, St. Louis, MI, USA). The pieces were incubated at 37 °C for 20 min, vortexed for 15 s, and transferred onto a 100 µm filter. The flowthrough contained the IELs. After repeating this step once, gut pieces were incubated in HBSS supplemented with 10 mM HEPES for 20 min, vortexed, and also put on a 100 µm filter. The flowthrough was spun down and transferred to a density gradient with 70% and 40% Percoll (GE Healthcare, Chicago, IL, USA) to separate the IEL-fraction. Afterward, lymphocytes were enriched by magnetical depletion with anti-EpCAM-, anti CD11c- and anti CD11b beads (all Miltenyi Biotech, Bergisch Gladbach, Germany). Purity was checked by flow cytometry and was routinely at a ratio of 20% CD4^+^ and 80% CD8^+^.

### 4.4. IEL—IEC Co-Culture

To establish IEL-IEC co-cultures, two-day-old organoid cultures were removed from matrigel, washed with cold PBS, and incubated with freshly isolated IELs in BCM at 37 °C for 30 min. Then, the cell suspension was spun down, resuspended in a 1:1 BCM-matrigel mix, and plated in a 48 well Nunclon Delta Surface plate (Thermofisher, Waltham, MA, USA) or ibidi plate (Ibidi, Gräfelfing, Germany). We usually calculated 100 two-day-old organoids along with 250.000 IELs per well. CCM media supplemented with 100 U/mL IL-2, 10 ng/mL IL-7 and 10 ng/mL IL-15 (all cytokines from Immunotools, Friesoythe, Germany) was added after matrigel polymerization. Co-cultures were split before imaging if indicated. To visualize IELs in the co-culture setup, IELs were labeled with a live cell dye (Cell Proliferation Dye eFluor 670, 1 µM, eBioscience, San Diego, CA, USA). Alternatively, IELs purified from *tdTOMATO^CD4/CD8^* reporter mice were used.

### 4.5. Flow Cytometry

For flow cytometry analysis of IELs from IEL-IEC co-cultures, the cells were washed out of the matrigel with cold PBS, spun down, and incubated with Cell Recovery Solution (Corning, New York, NY, USA) for 20 min on ice. Afterward, the cell suspension was incubated with pre-warmed TrypLE (Life Technologies, Carlsbad, CA, USA) for 5 min at 37 °C and vortexed afterward. This step was repeated three times. The single-cell suspension was washed with cold PBS and stained with rat anti-mouse CD3 (17A2, 1:100), hamster anti-mouse TCRβ (H57-597, 1:100), hamster anti-mouse TCRγδ (GL3, 1:200), rat anti-mouse CD4 (GK1.5, 1:200), rat anti-mouse CD8α (53–6.7, 1:200), rat anti-mouse CD8β (YTS156.7.7, 1:200), hamster anti-mouse CD103 (2E7, 1:100) and hamster anti-mouse CD69 (H1.2F3, 1:40) (all Biolegend, San Diego, CA, USA) in PBS supplemented with 3% FCS (FACS buffer) for 15 min. After washing the cell suspension with FACS buffer flow cytometry was carried out by LSR Fortessa (BD, Franklin Lakes, NJ, USA) and analyzed with FlowJo (version 10.6.2, BD, Franklin Lakes, NJ, USA).

### 4.6. Immunofluorescence

For immunofluorescent stainings, 5 µm small intestinal tissue slices were fixed with 2% paraformaldehyde (PFA) for 20 min at room temperature (RT), washed with phosphate-buffered saline (PBS), and blocked with histobuffer (10% FCS, 5% BSA in PBS) for 1 h at RT. The tissue was stained with anti-EpCAM AF488 (1:200, Biolegend, San Diego, CA, USA) and anti-CD3 AF647 (17A2, 1:100, Biolegend, San Diego, CA, USA) in histobuffer overnight at 4 °C. After washing, staining of nuclei was performed by a 5 min incubation in 1:10000 DAPI/Hoechst (Life Technologies, Carlsbad, CA, USA) at RT. After a washing step, the slices were mounted in fluorescence mounting media (Dako, Carpinteria, CA, USA) and stored at 4 °C until imaging.

To stain slices of embedded co-cultures, cell suspensions were resuspended in HistoGel (ThermoFisher Scientific), plated on a pre-cooled slide, and embedded in paraffin after HistoGel polymerization. 4 µm slices were cut and incubated at 60 °C for 30 min. Deparaffinization and rehydration were performed by applying Rothihistol (Carl Roth, Karlsruhe, Germany) for 5 min followed by 100%, 96%, and 70% ethanol for 5 min each. This procedure was repeated twice thereafter. For antigen retrieval, citrate buffer was used. Slides were stained with TSA-Kit (PerkinElmer, Waltham, MA, USA). For anti-CD3 staining, slides were incubated with hamster anti-mouse CD3 Biotin antibody (145-2C11, 1:100, Biolegend, San Diego, CA, USA) overnight at 4 °C. After a washing step, the tissue slices were incubated with Streptavidin diluted 1:100 at RT for 30 min followed by washing steps and incubation with 1:50 Cyanine-3-Tyramide reagent for 5 min. For the staining of nuclei, samples were incubated in 1:10000 DAPI/Hoechst (Life Technologies, Carlsbad, CA, USA) for 5 min. After washing, the slices were mounted in fluorescence mounting media (Dako, Carpinteria, Germany).

Samples for 3D imaging were also stained with TSA-Kit (PerkinElmer, Waltham, MA, USA) after 20 min fixation with 2% PFA. Here, the same protocol was used as described above except that co-cultures were plated and imaged on a 4 well ibidi plate (Ibidi, Gräfelfing, Germany).

### 4.7. Confocal Microscopy

Confocal fluorescent microscopy was performed with a Leica SP5 (Leica, Wetzlar, Germany). Slides were imaged upon Laser 405 nm, 488 nm, 561 nm, and 633 nm and PMT detection. A 20× magnification lens was used routinely.

### 4.8. 3D Visualization

3D reconstruction of co-culture samples was created after imaging with 5 µm z-stacks spacing using a Zeiss Spinning Disc Axio Observer Z1 (Zeiss, Oberkochen, Germany). IELs prestained with Cell Proliferation Dye eFluor 670 or purified from *tdTOMATO^CD4/CD8^* reporter mice were studied by confocal spinning disc microscopy, employing an EMCCD camera (Evolve 512 delta, Photometrics) at a magnification of 25×, with a band pass (BP) filter 629/62 and excited by a 561 nm Laser. IECs were imaged with a BP filter 525/50 and excited by a Laser at 488 nm. Z-stack data was analyzed with Fiji/ImageJ (version 1.53c, Wayne Rasband, National Institutes of Health, USA), imported with Bio-Formats [33], and rendered with 3Dscript [34].

### 4.9. Live Imaging

For confocal live cell imaging, Zeiss Spinning Disc Axio Observer Z1 live cell observer at the Optical Imaging Centre Erlangen (OICE) was used routinely as specified in the figure legends. Co-cultures were plated on a 4 well ibidi plate and incubated in a 37 °C, 5% CO_2_ chamber during the imaging process. A 25× magnification objective was used. Organoids were imaged in differential interference contrast (DIC) illumination. The pre-stained IELs were imaged via Evolve camera, BP filter 690/50, and excited by Laser 635. For Lucifer Yellow Assay, Lucifer Yellow CH dilithium salt fluorescent stain (Sigma, St. Louis, MI, USA) was added 1:500 to the culture 30 min before imaging. For this assay bright field illumination was turned off to image organoids in black and surrounding area imaged via Evolve camera, BP filter 525/50 and excited by Laser 488. Time-lapse experiments were performed by taking a picture every 30 s for up to 45 min. Time-lapse data was analyzed with Fiji/ImageJ (version 1.53c, Wayne Rasband, National Institutes of Health, USA) and imported with Bio-Formats [33]. Cells tracks were computed with TrackMate [35], using the LoG (Laplacian of Gaussian) detector with an estimated blob diameter of 7 micron and a threshold of 80 and the Simple LAP tracker (Linear Assignment Problem) with linking max distance and gap-closing max distance both set to 15 microns and gap-closing max frame gap set to 2.

Some live cell studies were visualized by confocal fluorescent microscopy using a Leica SP5 at the First Department of Medicine, Universitätsklinikum Erlangen. Organoids were imaged in DIC illumination. IELs pre-stained with 1 µM Cell Proliferation Dye eFluor 670 (eBioscience, San Diego, CA, USA) were imaged with Laser 631 and a photomultiplier tube (PMT) detector. A 20× magnification was used.

### 4.10. Characterization of IEL mobility

A track is defined as a contiguous, comprehensible movement of one cell over several frames and is represented by a single yellow line. It ends if the cell is not visible for more than two frames or if it is apart more than 15 microns in consecutive frames. IEL tracks were analyzed in the whole area or in separated regions inside and outside of organoids. The inside and outside of organoids were defined by freehand selection.

A transition event is defined as a track that crosses from inside to outside or vice versa.

The speed of a cell represents the velocity of an IEL in the co-culture and was calculated by measuring the horizontal distance, represented by the track length that a cell passes over time during life cell imaging. Tracks with a speed below 0.05 micron per second were excluded to adjust for limitations resulting from the transformation of a 3D model into 2D measurements.

Displacement represents the length of an individual track. The maximum displacement was defined as the mean of the top 15 displacement values per time-lapse movie.

For standardization purposes in comparative mobility studies, all co-cultures were performed routinely with two days old organoids, incubated overnight before imaging, and analyzed using an objective lens at 25× magnification with a consistent field of view. Transitions were counted by a self-written Fiji macro analyzing the output of Trackmate.

### 4.11. Statistical Analysis

For the comparison of means of two groups with assumed normal distribution, an unpaired two-tailed Student’s t-test was applied. Graphpad Prism software (version 9.0.2, Graphpad Software, San Diego, CA, USA) was used for statistical analysis.

## Figures and Tables

**Figure 1 ijms-22-05148-f001:**
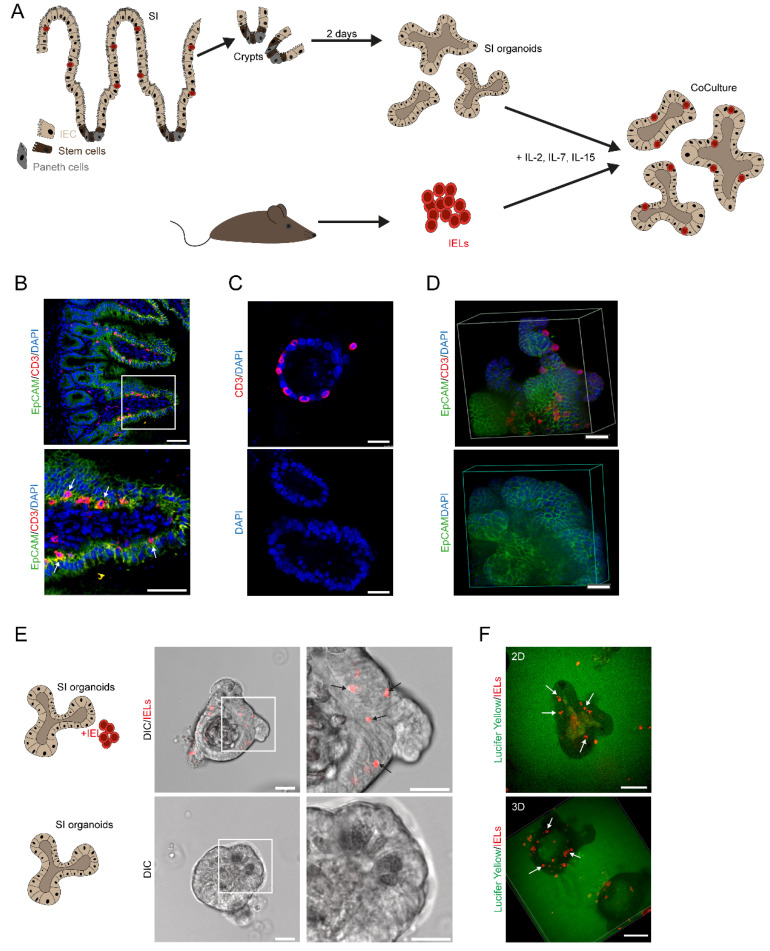
An experimental ex vivo co-culture system resembles the spatial distribution of IELs within the IEC layer. (**A**) Graphical abstract of the IEL-IEC co-culture setup. Crypts were isolated from the small intestine and cultured for two days to form organoids. IELs were freshly purified on day two, added to the organoids, and supplemented with IL-2, IL-7, IL-15. (**B**) Ileal tissue sections from unchallenged 12 weeks old wildtype mice were stained with immunofluorescently labeled antibodies anti-CD3 (red), anti-EpCAM (green), and DAPI (blue) and analyzed by SP5 confocal microscopy. Arrows indicate the localization of IELs. Scale bar 50 µm. (**C**) IEL-IEC co-cultures were embedded in paraffin, stained with immunofluorescently labeled anti-CD3 antibody (red), counterstained with DAPI (blue), and imaged by SP5 confocal microscopy. Scale bar 25 µm. (**D**) IEL-IEC co-cultures were plated on ibidi plate and after fixation stained with anti-EpCAM (green), anti-CD3 (red), DAPI (blue) and analyzed by spinning disc microscopy and 3D reconstruction. Scale bar 15 µm. (**E**) IEL-IEC co-cultures with pre-stained IELs (Cell Proliferation Dye, red) were studied by live cell imaging using confocal microscopy in differential interference contrast (DIC). Scale bar 50 µm. (**F**) IEL-IEC co-cultures with pre-stained IELs (Cell Proliferation Dye, red) and matrigel background staining with Lucifer Yellow (green) were analyzed by live cell imaging using spinning disc microscopy including 3D reconstruction as indicated. Scale bar 25 µm. Arrows highlight the localization of IELs. All co-cultures shown in Figure 1 were usually split before imaging. Data from B–E are representative of at least three independent experiments.

**Figure 2 ijms-22-05148-f002:**
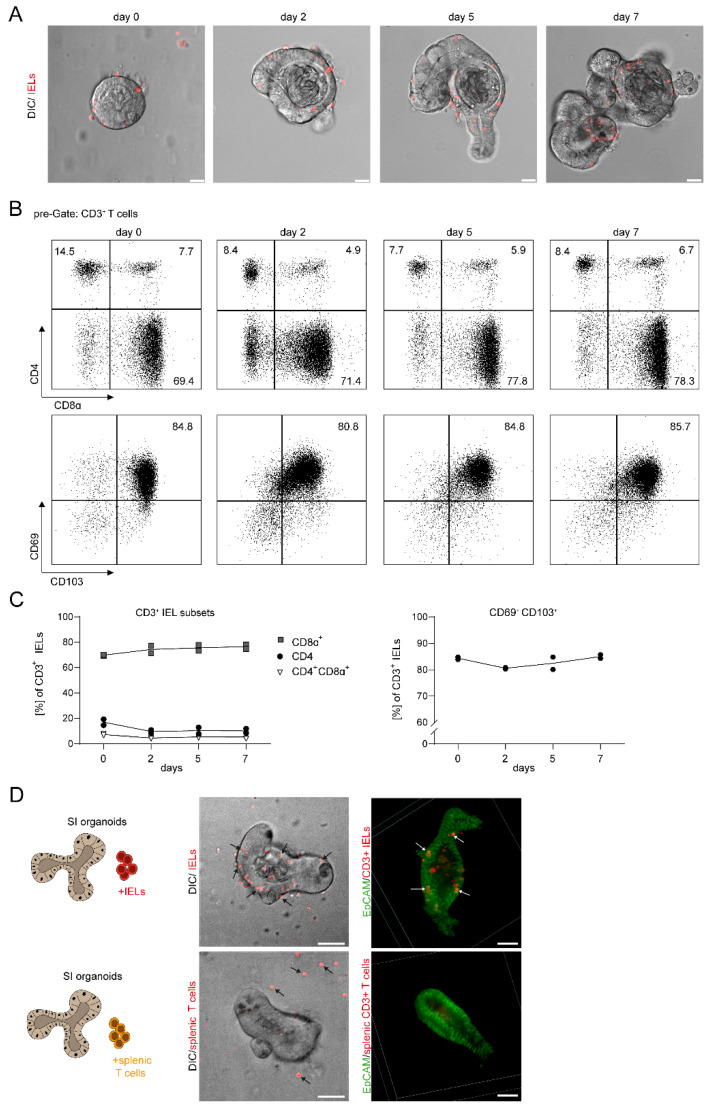
IELs remain alive, phenotypically stable, and localize to the IEC layer during a prolonged period of ex vivo co-culture. (**A**) IEL-IEC co-cultures with fluorescently pre-stained IELs (Cell Proliferation Dye, red) were monitored by confocal microscopy by SP5 in DIC from day 0 to day 7. Co-cultures were split before imaging. Scale bar 25 µm. Representative images of at least three independent experiments are shown. (**B**) IELs were removed from IEL-IEC co-cultures at different time points as indicated, stained with fluorescently labeled antibodies directed against CD3, CD4, CD8α, CD69, and CD103, and analyzed by flow cytometry. Representative FACS blots of two independent experiments are shown. (**C**) FACS-based kinetic analysis is shown for the expression and ratio of selected surface markers on CD3^+^ IELs obtained from IEL-IEC co-cultures as indicated. Results are pooled from two independent experiments. (**D**) Co-cultures were set up with small intestinal IEC organoids and either IELs or splenocytes from *tdTOMATO^CD4/CD8^* reporter mice (red) as indicated. Co-cultures were analyzed by live cell imaging with confocal microscopy in DIC with the spinning disc (left) or after fixation and IF staining for EpCAM (right) with T cells from *tdTOMATO^CD4/CD8^* mice. IELs and splenic T cells are shown in red. Arrows point to typical localizations of IELs and splenocytes, respectively. Scale bar 50 µm. Co-cultures with T cells from *tdTOMATO^CD4/CD8^* reporter mice were incubated overnight before imaging. Representative images of three independent experiments are shown.

**Figure 3 ijms-22-05148-f003:**
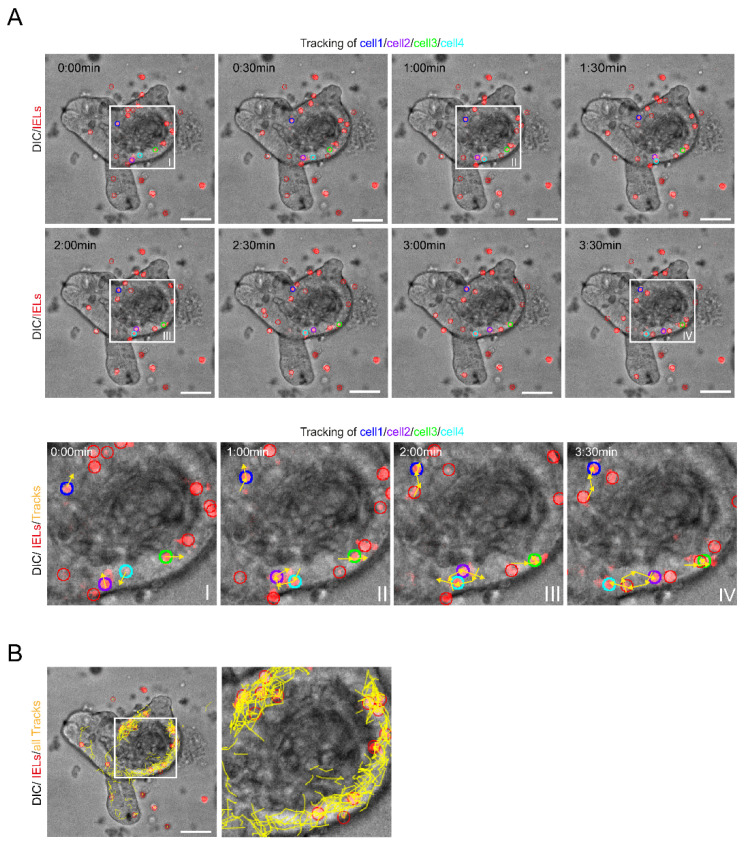
The mobility of IELs within the IEC layer can be monitored by life cell imaging and tracked with time-lapse studies. (**A**) IEL-IEC co-cultures with pre-stained IELs (Cell Proliferation Dye, red) were serially monitored by time-lapse imaging using spinning disc microscopy with 25× objective magnification in DIC. Time series demonstrates eight frames with each frame taken 30 sec apart. Four exemplary IELs are marked in blue, purple, green, and turquoise, respectively. The movements of the four exemplary IELs within the epithelium during the observation period are highlighted by yellow arrows and lines (tracks) in the close-up views (I–IV). Scale bars 50 µm. (**B**) Tracks of all IELs detectable during the observation period of 45 min are shown by yellow lines. Scale bar 50 µm. Co-culture was incubated overnight before imaging.

**Figure 4 ijms-22-05148-f004:**
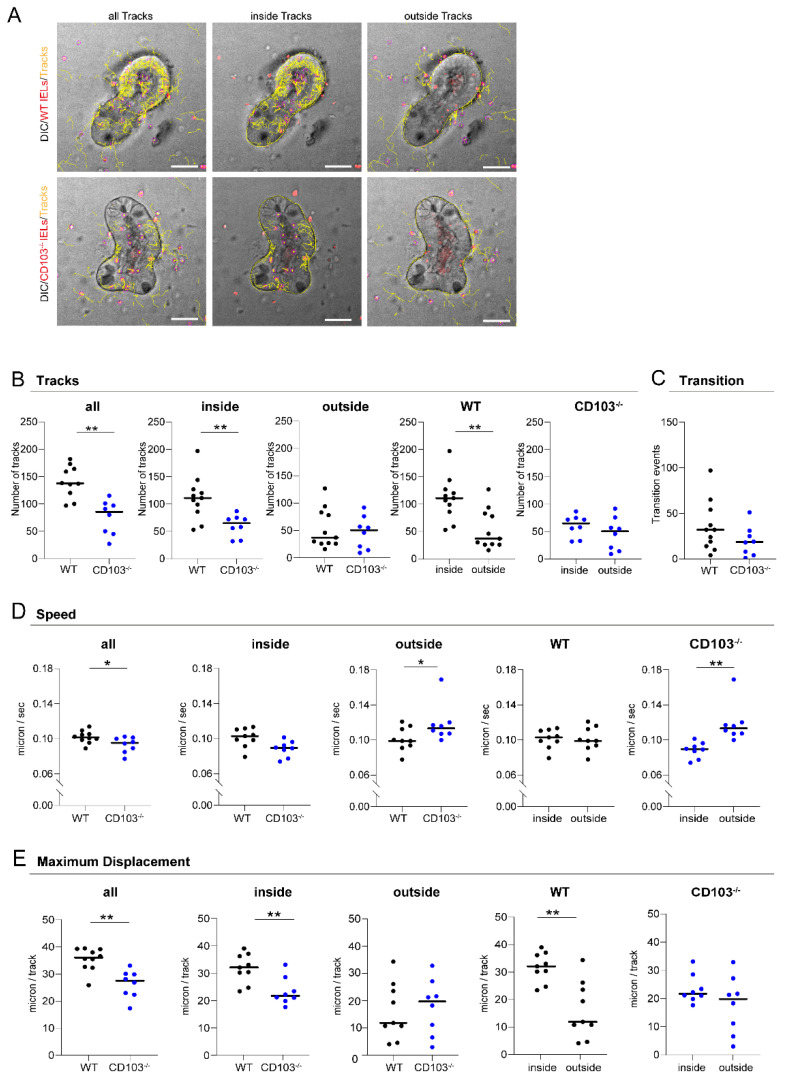
CD103-deficient IELs display aberrant mobility and migration patterns. (**A**) IEL-IEC co-cultures with two days old organoids and fluorescently labeled IELs (Cell Proliferation Dye, red) from wildtype (WT) and *CD103^−/−^* mice (as indicated) were serially studied after overnight incubation by time-lapse imaging using spinning disc microscopy in differential interference contrast (DIC) with an objective lens at 25× magnification. IEL tracks of the entire area (all tracks, left), within IECs (inside tracks, middle), or around IECs (outside tracks, right) are marked in yellow. Scale bar 50 µm. (**B**) The number of IEL tracks within the subgroups as specified were quantified in IEL-IEC co-cultures with IELs from wildtype and *CD103^−/−^* mice, respectively. A track is a contiguous, comprehensible movement of one cell over several frames and is represented by a single yellow line (see Section 4.10). (**C**) The number of IEL tracks occurring as transition events, i.e., crossing between inside and outside of IEC organoids, was analyzed in IEL-IEC co-cultures with IELs from wildtype and *CD103^−/−^* mice, respectively. (**D**) The mean speed of cells including within the subgroups as specified was calculated in IEL-IEC co-cultures with IELs from wildtype and *CD103^−/−^* mice, respectively. (**E**) The maximum displacement, i.e., mean of the highest 15 track length values per time-lapse movie, was compared in IEL-IEC co-cultures between IELs from wildtype and *CD103^−/−^* mice, respectively. Please see Section 4. for the definition of terms and more details on methodology. Co-cultures were incubated overnight before imaging. *n* = 8–10 per group. All data are representative of at least three independent experiments. * *p* < 0.05, ** *p* < 0.01.

## Data Availability

Data is contained within the article or Appendix A.

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
