# Peer review of "Dynamic Imaging of IEL-IEC Co-Cultures Allows for Quantification of CD103-Dependent T Cell Migration"

_ijms, 2021, doi:10.3390/ijms22105148_

Round 1
Reviewer 1 Report
This is a very interesting paper building on the work by the Nozaki group and the paper's authors, establishing and characterising an ex vivo model to investigate the relationship between intraepithelial lymphocytes and small intestine epithelial cells. The paper is well written with a couple of grammar errors; line 23 life should be replaced with live, line 405 remove In.
The results are clear and well laid out however, the data in Figure 2 C, CD3+ IEL subsets should be displayed as experimental numbers not pooled, as was done in the CD69+ CD105+ graph.
Could the authors explain in the paper why they stained the IELs with cell proliferation dye in the live cell imaging and confocal microscopy experiments instead of anti CD3?
Despite using a very similar ex vivo set up to the Nozaki group the authors did not see a noticeable change in IEL numbers after 7 days incubation. I think the authors should give an explanation as to why their results differed from the published data.
Of note, the group stated that, for the first time they showed the better migratory ability of the IELs compared to splenic T cells. However Figure 2 D does not show splenic T cells in the IF picture, not even in the extra organoid space, leading the reader to question whether the splenic T cells were added to the organoid co-culture.
The authors clearly layout the limitation of the method, in that only the interaction between 2 cell types are investigated, whereas in vivo multiple cell types and micro-organisms are involved.
Reviewer 2 Report
This is a very interesting work that focuses on an important issue with plausible implications on intestine research, particularly in IBD.
Questions to the authors:
What is your explanation that the movement in CD103-/- the movement is significantly high outside than inside? Figure 4D
What was the longest time you monitor IELs?
Whit time can be possible that outside IEL enter inside or vice versa?
Have you tried to work with ILEs that were not cultured before with a cocktail including IL-2, IL-7, and IL-15?
